# Prevalence of Chronic Diseases and Activity-Limiting Disability among Roma and Non-Roma People: A Cross-Sectional, Census-Based Investigation

**DOI:** 10.3390/ijerph16193620

**Published:** 2019-09-26

**Authors:** Ferenc Vincze, Anett Földvári, Anita Pálinkás, Valéria Sipos, Eszter Anna Janka, Róza Ádány, János Sándor

**Affiliations:** 1Department of Preventive Medicine, Faculty of Public Health, University of Debrecen, Kassai St 26/B, H-4028 Debrecen, Hungary; vincze.ferenc@sph.unideb.hu (F.V.); anettefoldvari@gmail.com (A.F.); palinkas.anita@sph.unideb.hu (A.P.); sipos.valeria@sph.unideb.hu (V.S.); 2Doctoral School of Health Sciences, University of Debrecen, Kassai St 26/B, H-4028 Debrecen, Hungary; 3Department of Dermatology, Faculty of Medicine, University of Debrecen, H-4032 Debrecen, Hungary; janka.eszter.a@gmail.com; 4WHO Collaborating Centre on Vulnerability and Health, Public Health Research Institute, University of Debrecen, Kassai St 26/B, H-4028 Debrecen, Hungary; adany.roza@sph.unideb.hu; 5Public Health Research Institute, University of Debrecen, Kassai St 26/B, H-4028 Debrecen, Hungary; 6MTA-DE-Public Health Research Group, Public Health Research Institute, University of Debrecen, Kassai St 26/B, H-4028 Debrecen, Hungary

**Keywords:** Roma minority, health status, census, risk assessment, impact assessment, monitoring

## Abstract

The lack of recommended design for Roma health-monitoring hinders the interventions to improve the health status of this ethnic minority. We aim to describe the riskiness of Roma ethnicity using census-derived data and to demonstrate the value of census for monitoring the Roma to non-Roma gap. This study investigated the self-declared occurrence of at least one chronic disease and the existence of activity limitations among subjects with chronic disease by the database of the 2011 Hungarian Census. Risks were assessed by odds ratios (OR) and 95% confidence intervals (95% CI) from logistic regression analyses controlled for sociodemographic factors. Roma ethnicity is a risk factor for chronic diseases (OR = 1.17; 95% CI: 1.16–1.18) and for activity limitation in everyday life activities (OR = 1.20; 95% CI: 1.17–1.23), learning-working (OR = 1.24; 95% CI: 1.21–1.27), family life (OR = 1.22; 95% CI: 1.16–1.28), and transport (OR = 1.03; 95% CI: 1.01–1.06). The population-level impact of Roma ethnicity was 0.39% (95% CI: 0.37–0.41) for chronic diseases and varied between 0 and 1.19% for activity limitations. Our investigations demonstrated that (1) the Roma ethnicity is a distinct risk factor with significant population level impact for chronic disease occurrence accompanied with prognosis worsening influence, and that (2) the census can improve the Roma health-monitoring system, primarily by assessing the population level impact.

## 1. Introduction

The Roma population is the largest ethnic minority group in Europe. The common opinion is that the health status of this population is poor compared with that of the non-Roma population. Recently, this rather anecdotal belief has been convincingly supported by a growing number of quantitative investigations. The current urgency for intervention is hardly debatable [1], but effective interventions are hindered by our limited knowledge regarding the details of the mechanisms that have led to the substantial population-level health deficit among the Roma population [2,3,4].

It has been thoroughly demonstrated that Roma people are overrepresented in the marginalized communities, living in relatively unhealthy environments, and have an unhealthier lifestyle than the non-Roma population [5,6,7,8,9,10]. In addition, the access of Roma people to care is impaired, leading to high levels of unmet health needs [4,11,12,13,14,15,16,17,18]. It is yet to be clarified whether the Roma people themselves are a distinct intervention target or whether the unhealthy environment, lifestyle, and restricted availability of health services, to which many non-Roma people are exposed, should be targeted by organized interventions. In brief, is the Roma ethnicity an important and distinct risk factor to be targeted by population-level interventions?

Furthermore, it is obvious that the Roma people are not a homogeneous social stratum. Our knowledge of the role of this heterogeneity in determining the health status of this population is quite limited. Moreover, we do not know whether it is justified at all to use this dichotomization (distinguishing Roma and non-Roma persons) in epidemiological investigations of the risk associated with Roma ethnicity [19,20,21]. 

It seems that higher quality publications on the health impact of Roma ethnicity use more narrow case definitions or more specific settings [22,23,24,25,26]. This approach results in knowledge that relies mainly on subgroup specific findings [7,10,12,25], observations that cannot be extrapolated to the whole Roma population [10,27] and limited knowledge of the public health importance of Roma ethnicity as a risk factor.

All of these problems are reflected in the questionable effectiveness of Roma-targeted health interventions [3,4,28,29], showing that program elaboration, monitoring, and evaluation require the development of currently applied epidemiological methods. At present, responsible policy-makers face a scarcity of useful data on the population-level health impact of Roma ethnicity.

Realizing the weight of this challenge, the European Union accepted “An EU Framework for National Roma Integration Strategies up to 2020” (NRIS) in 2011. It commits Member States to integrate policies that focus on Roma minorities in a “clear and specific” way and address the requirements of Roma minority with “explicit actions” in order to reduce the health gap between Roma and non-Roma populations. [30] Furthermore, the NRIS implementation-monitoring states that achieving the goals of the national adaptation of NRIS [31] without defining measurable indicators and operating a robust monitoring system is doubtful [32,33].

Unfortunately, there is no recommended, detailed design for health-monitoring of the Roma population in NRIS that could solve the two interrelated issues of epidemiological monitoring: identification of Roma individuals and definition of sampling frame. Although, there are initiatives to overcome these methodological shortcomings [34,35]. In practice, the combined efforts by subgroup investigations related to the mechanisms connected to Roma ethnicity, which are completed with self-declared ethnicity-based population-level investigations, are applied to obtain useful Roma health-specific data. Taking into account that surveys are inevitably subject to sampling bias because of the nonexistence of a Roma sampling frame (e.g., the recently conducted Second European Union Minorities and Discrimination Survey study coordinated by the European Union Agency for Fundamental Rights has representativity limitations arising from the sampling frame) [36], a census that would provide data representative of the population could be a useful part of monitoring.

Self-declared ethnicity-related data based on national censuses have been available for decades in many countries [37,38,39,40]. To avoid the well-known under-registration of Roma ethnicity, multiple questions used to be applied in census questionnaires. Governments ensure the regularity and the feasibility of censuses required for monitoring. If a census collects data on health status and ethnicity along with the most important sociodemographic risk factors of health impairments, then it can establish a basis for comparative investigations of Roma and non-Roma investigations regularly and in a feasible manner.

Our aim is (1) to compare the occurrence of chronic diseases and activity limitations among the Roma and non-Roma populations in order to describe the riskiness and impact of Roma ethnicity independently from sociodemographic factors using census-derived data and (2) to describe the added value of a census-based, obviously not perfect methodology applying in itself, to the monitoring of the health gap between Roma and non-Roma populations.

## 2. Materials and Methods 

### 2.1. Setting

This study investigated an anonymized database of the last Hungarian national census conducted in 2011. The cross-sectional survey covered the whole Hungarian population. The data collection was carried out from 1 October 2011 to 31 October 2011. The study population included all Hungarian citizens living in the country or staying temporarily abroad for a period of less than 12 months as well as all foreign citizens and stateless persons living in Hungary for a period of more than 3 months.

The availability of the database was provided by the Hungarian Central Statistical Office (HCSO). The study was approved and supervised by the institutional board of the HCSO, which is responsible for both the utilization of the census database and the protection of human rights in handling sensitive personal data (KSH/ADKI/1156/2014).

### 2.2. Explanatory Variables and Outcome Measures

The first part of the census questionnaire [41] was focused on housing conditions (type of walls and presence of utilities, such as bathroom, flush toilet, electricity, water and hot water supply, and heating system). Because the response to these questions was compulsory, apart from homeless people and those who are living in institutions, all subjects’ living conditions were characterized by these parameters. All factors were used as proxy indicators of deprivation in the analysis.

The second part of the census questionnaire was focused on personal characteristics [42]. The responses were compulsory in this part of the questionnaire as well, apart from voluntary questions related to ethnicity, the presence of chronic disease, and activity limitation. Subjects were classified according to covariate variables, such as sex, age, marital status, highest level of education, and employment status. The following age groups were used: 0–5, 6–17, 18–34, 35–59, 60–64, and 65+ years. The highest level of education was categorized as less than primary, primary, vocational, high school, and tertiary school. Those individuals less than 6 years old or attending an educational institution were classified as not having completed their education. Employment status was described as working, unemployed, retired, receiving social benefits, dependent, or student. Single, coupled, divorced, and widowed were distinguished with respect to marital status.

Four questions referred to self-declared ethnicity. These questions included primary self-declared ethnicity (“Which ethnicity do you feel you belong to?”), secondary self-declared ethnicity (“Do you think you belong to another ethnicity too?”), first language (“What is your first language?”), and secondary language (“In what language do you usually speak with family members or friends?”). Those who responded with “Roma” to any of these questions were regarded as Roma people. All others were considered to be non-Roma people.

The primary outcome variable was the self-declared occurrence of at least one chronic disease. The type of disease was not requested. The existence of activity limitations among subjects with chronic disease was investigated as a secondary outcome. The activity limitations were sub-grouped according to the affected function: self-sufficiency, everyday life, learning-working, family life, transport, communication and community life.

### 2.3. Statistical Analysis

First, crude prevalence of having at least one chronic disease and activity limitations among those with chronic disease were calculated for the Roma and non-Roma subpopulations. The association with the Roma ethnicity was evaluated by χ^2^-test.

Then, internal indirect standardization was used to control for the confounding effect of age, sex, and level of education. The age- and sex-standardized prevalence ratios (SPR) and age-, sex-, and education-standardized prevalence ratios (SPR_e_) were calculated both for the Roma and non-Roma populations. Next, we assessed the risk associated with Roma ethnicity by the ratio of Roma-specific and non-Roma-specific standardized prevalence ratios (RR = SPR_Roma_/SPR_non-Roma_; RR_e_ = SPR_e,Roma_/SPR_e,non-Roma_). The 95% confidence intervals (95% CI) of the calculated measures were used in the statistical evaluation.

Then, we used multiple logistic regression modeling to investigate the influence of Roma ethnicity, independent of housing conditions (wall quality, public utilities for bathroom, flush toilet, electricity, water and hot water supply, and heating system) and personal characteristics (sex, age, level of education, employment, and marital status). Associations were quantified by odds ratios (OR) and corresponding 95% CIs.

Finally, point estimates and 95% CIs of the relative risks estimated by standardization or logistic modeling were used to describe the impact of Roma ethnicity on chronic disease and the occurrence of activity limitation. Attributable risk fractions among Roma (AR_Roma_) and the whole population (AR_Population_) were computed with corresponding 95% CIs.

Statistical computation was performed with STATA 14.0 (Stata Corporation, College Station, TX) software.

## 3. Results

There were 9,937,628 subjects who participated in the Hungarian census in 2011. The restriction of this population to the Hungarian citizens living in Hungary resulted in a database of 9,794,318 persons. Because the response rate for ethnicity and chronic disease-related questions was 78.36%, the database that could be used to analyze the determinants of chronic disease contained 7,674,607 records. In the case of logistic regression modeling, 166,366 records were excluded because the housing conditions data was not available. This exclusion reduced the response rate to 76.66%. On the basis of self-declarations, 3.83% (294,189 persons) were classified as Roma individuals, and 21.19% (1,626,447 persons) had at least one chronic disease in the study population.

The response rate for the question about activity limitation among those with chronic disease was 87.09%. This resulted in a database of 1,416,424 persons with chronic disease and who reported activity limitations (in the multivariate modeling, 40,763 whose housing conditions data was not available were also excluded, resulting in an 84.58% response rate for that analysis). There were 38,800 (2.74%) Roma persons in this database (Figure 1).

The sex representations among the Roma and non-Roma people were significantly different because of the higher proportion of males among the Roma population. The Roma age structure deviated significantly from the non-Roma age composition. The Roma were overrepresented in younger age groups and underrepresented among the older age groups. The level of education was much lower among Roma than among non-Roma. The percentage of individuals who had primary or less than primary education was 53.31% among the Roma population and 23.84% in the non-Roma population (Table 1). All of the other personal and housing-related deprivation indicators applied in the multiple logistic regression models indicated the poorer status of the Roma population in comparison to the non-Roma population (Table A1).

### 3.1. Chronic Disease Occurrence

The crude prevalence of having at least one chronic disease was significantly lower in the Roma population (14.75%) than in the non-Roma population (21.45%) (Table 2). However, significant risk elevation was observed among the Roma population after adjusting for age and sex (RR = 1.41) and after adjusting for age, sex, and education (RR_e_ = 1.11). We found that the Roma ethnicity was a significant risk factor for having at least one chronic disease in the multivariate logistic model that controlled for deprivation indices (OR = 1.17) (Table 3).

Each of the age- and sex-adjusted; age-, sex-, and education-adjusted; and multiply adjusted estimations showed a significant impact of Roma ethnicity on chronic disease occurrence, both within the Roma population and in the whole population. (Table 4)

### 3.2. Activity Limitation among Subjects with Chronic Disease

Table 2 summarizes the unadjusted frequencies of activity limitations among subjects with chronic disease by the studied functions. Each outcome was significantly more frequent in the Roma population compared to the non-Roma population.

Age- and sex-standardized measures showed a similar pattern. The self-sufficiency limitation was more frequent in the Roma population (RR = 1.75). The activity limitation in everyday life showed significantly higher frequency in Roma (RR = 1.64). The Roma ethnicity was also associated with increased risk of limitation among the Roma population in the fields of learning-working (RR = 1.59), family life (RR = 1.60), transport (RR = 1.54), communication (RR = 1.44), and community life (RR = 1.14). Each age- and sex-adjusted risk ratio and attributable fraction was highly significant. (Table 3 and Table 4)

According to the SPR_e_ for activity limitations, there was no evidence of differences between the Roma and non-Roma populations with respect to self-sufficiency or community life-related activities. Significant risk elevation was observed in the fields of everyday life activities (RR_e_ = 1.18, AR_Roma_ = 15.25; AR_Population_ = 0.53), learning-working (RR_e_ = 1.19, AR_Roma_ = 16.97; AR_Population_ = 0.98), family life (RR_e_ = 1.22, AR_Roma_ = 18.03; AR_Population_ = 0.89), and transport (RR_e_ = 1.16, AR_Roma_ = 13.79; AR_Population_ = 0.39) among Roma with chronic disease. The activity limitation in communication was significantly less frequent among Roma people (RR_e_ = 0.86, AR_Roma_ = −16.28; AR_Population_ = −0.64) (Table 3 and Table 4).

In the multiple logistic regression analyses, Roma ethnicity was associated with higher risk of activity limitation in the fields of everyday life activities (OR = 1.20, AR_Roma_ = 16.67; AR_Population_ = 0.58), learning-working (OR = 1.24, AR_Roma_ = 19.35; AR_Population_ = 1.19), family life (OR = 1.22, AR_Roma_ = 18.03; AR_Population_ = 0.89), and transport (OR = 1.03, AR_Roma_ = 2.91; AR_Population_ = 0.08). There was no significant difference in the activity limitation between the Roma and the non-Roma populations in the fields of self-sufficiency, communication, and community life (Table 3 and Table 4).

## 4. Discussion

### 4.1. Main Findings

The crude prevalence of chronic diseases among the Roma population was remarkably lower compared to that among non-Roma. The opposite difference was observed for each studied activity limitation, which can be attributed, theoretically, to the slower disease progression, resulting in fewer complications or to faster prognosis, resulting in early death. In light of the published results from many settings on the worse disease prognosis for Roma populations [14,17] and some data on the elevated mortality risk among Roma populations [16,17], the latter explanation seems to be more likely.

The analyses controlled for the younger age structure of the Roma population, with age and sex adjustment, demonstrating that Roma people have an elevated risk both for chronic disease occurrence and for each studied activity limitation. These results suggest that both disease development and prognosis among those with disease is faster among the Roma population.

After standardization by level of education, the risk of chronic disease was significantly mitigated but remained significant. This result shows that a considerable part of the age- and sex-adjusted excess risk was partly attributable to the poor education status of the Roma people. This result is in accordance with the findings of higher disease risk among the Roma populations [14,18,24,43,44]. The elevated age-, sex-, and education-corrected risk of activity limitations proved to be significant in the field of everyday life, learning-working, family life, and transport. While in the field of communication, this risk proved to be reduced among Roma people. This result was also consistent with previously published results [45,46]. Self-sufficiency and community life showed no difference among Roma and non-Roma patients with chronic disease.

The more extensive adjustment by multiple regression modeling confirmed the findings achieved by age, sex, and education adjustment. The only qualitative difference was that the activity limitation in communication was not significant in the regression modeling. Two quantitative modifications were also observed. The risk estimation was significantly higher for chronic disease occurrence and was significantly decreased for activity limitations in transport in the regression models.

On the basis of the multiple regression modeling, the impact of Roma ethnicity on chronic disease occurrence was 14.53%, showing that the Roma ethnicity is a distinct and important risk factor. On the other hand, the Roma ethnicity did not prove to be a risk factor among Roma people with chronic disease with regard to each of the studied activity limitations. In different fields, the impact varied between 0 and 19.35%.

The estimated population-level impact of 0.39% for chronic diseases and between 0 and 1.19% for activity limitations is modest at first glance. Taking into consideration that the Roma population is considerably underestimated by the census, for the actual size of the Roma population may be 870,000 as estimated by a Hungarian study focused on the estimation of the real population size using external Roma identifiers based on active contributions of the local governments [47], the corrected AR_Population_ for chronic disease risk is 1.16% (95% CI: 1.10–1.22). Although this measure is not directly comparable to the percentage of total loss of disability-adjusted life years (DALY), since the consequences of chronic diseases are variable, leading to premature death and functional impairments, this AR_Population_ is in the same range as the percentage of total DALY from road injuries (1.52%), breast cancer (1.30%), cardiomyopathy and myocarditis (1.24%), and alcohol use disorders (1.04%).

### 4.2. Strengths and Limitations

The main strength of this census-based investigation is that it covered the whole country, thus avoiding selection bias via preparation of a sampling frame. On the other hand, the census database comprised of nearly 10 million respondents, ensuring high statistical power for the analyses.

The achieved response rates (78.36% for chronic disease, 87.09% for activity limitation among persons with chronic disease) were relatively high compared with those observed in the Hungarian implementations of the European Health Interview Surveys (72% in 2009 and 62% 2014), thus resulting in smaller selection bias in the census-based evaluation than in the surveys.

However, this investigation could not avoid self-report-derived ethnicity misclassification and the remarkable underestimation of the Roma population’s size. Although this validity issue has to be acknowledged [48,49,50], the results from parallel applications of self-report and interviewer classifications in health surveys suggest that this validity issue has minor importance in the investigation of population-level health status differences between Roma and non-Roma people. It is a likely consequence of the fact that the self-declared Roma subjects tended to live in more segregated circumstances, have worse health status, and be less educated than those that did not declare Roma identity [21,51].

There was a misclassification with respect to the studied outcomes because of self-declaration. Because health status was not described in detail, the importance of this potential bias may be modest. On the other hand, low specificity of the outcome classification restricts the usefulness of census-derived data in elaborating and monitoring interventions, of course.

Our efforts to control for confounding effects was far from complete in this investigation. There were factors (e.g., the fact that income is difficult to capture in census settings) that should have been considered in the adjustment for sociodemographic status. Further bias was elicited by the lack of data on health services provided for participants. The variable availability and quality of care leads to variable perceptions of disease and activity limitations.

Because of the cross-sectional nature of the census, the elevated risks of activity limitations among persons with chronic disease cannot be interpreted as evidence for worse prognoses associated with chronic disease among Roma.

### 4.3. Implications

Taking into account that (1) the census can estimate the health impact of the Roma ethnicity among the Roma population and within the whole population, (2) it is enough to use the census-collected data to obtain reliable sociodemographic factors to adjust the estimations of Roma ethnicity-related risks, (3) the self-report-derived misclassification, which needs further investigations, seems not to be strong enough to prevent the application of census-derived findings in intervention planning and evaluation, the addition of the census to public health monitoring of the health of the Roma population seems to be justified.

However, the only available census-based evaluations are not able to meet the criteria of public health monitoring, mainly because the health status assessment is not detailed enough to provide decision-makers with an identifiable intervention target. Only the magnitude of Roma ethnicity-related health problems and the potential health gain that can be achieved by effective interventions among Roma can be identified by the census. While the details of such interventions can be ensured by population surveys and setting specific epidemiological investigations.

The proposal for the more intensive use of census data in health monitoring of the Roma population can be supported by the success of policies to reduce ethnic/racial inequalities in the United States and in the United Kingdom, which use census-derived ethnicity- and race-specific data in establishing and evaluating interventions [37,38,40].

## 5. Conclusions

Our investigations demonstrated that the Roma ethnicity is a distinct risk factor independent of sociodemographic status. It suggests significant impact among the Roma population for chronic disease occurrence accompanied with prognosis worsening influence on activity limitation in everyday life, learning-working, family life, and transport but not in self-sufficiency, communication, and community life. It is estimated that 1.16% of chronic diseases can be attributed to the Roma ethnicity in the whole population. Although, census data cannot determine in detail the targets for interventions, they can be used to estimate the general importance of the Roma ethnicity-related problems and the urgency of intervention, which is not achievable with population-level surveys and setting specific epidemiological investigations, according to previous experience. Census based impact assessment of Roma ethnicity on chronic disease occurrence of 1.16% was not described formerly neither by analytic investigations nor households surveys. Since sociodemographic risk adjustment can be carried out using census data, the census can provide useful data in spite of the uncertainties on account of self-declared ethnicity assessments. Altogether, the census, in addition to population-level surveys and subgroup-specific epidemiologic investigations, can significantly improve the health-monitoring system of the Roma population.

## Figures and Tables

**Figure 1 ijerph-16-03620-f001:**
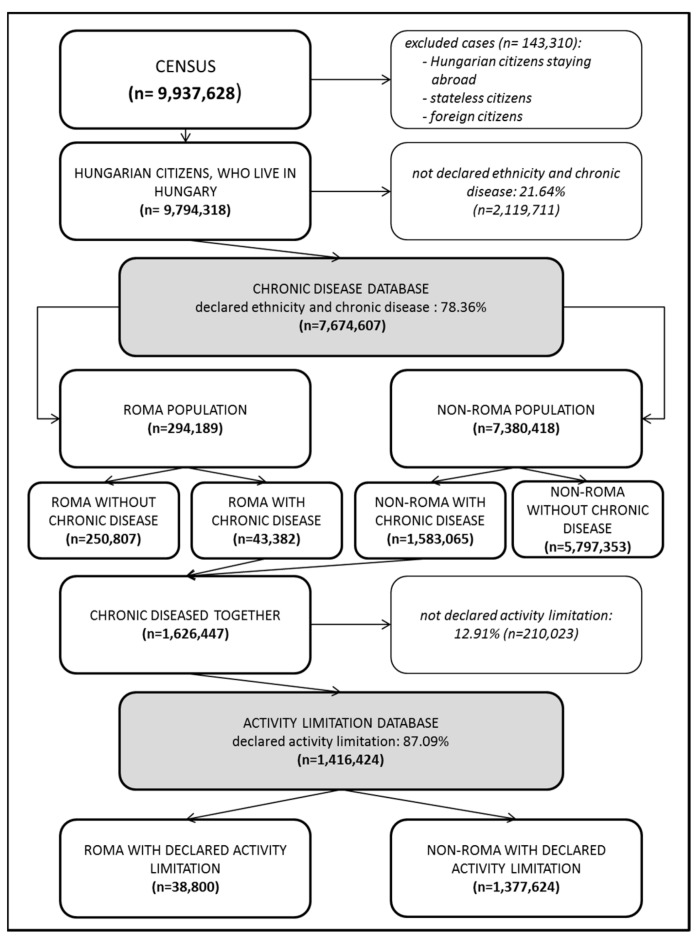
Studied groups’ selection from Hungarian census 2011 population.

**Table 1 ijerph-16-03620-t001:** Demographic characteristics of the study population and the differences between the Roma and non-Roma population evaluated by χ^2^-test.

	Total (%)	Ethnicity		*p*-Value
		Roma (%)	Non-Roma (%)	
Sex				
Male	3,634,387 (47.36%)	148,889 (50.61%)	3,485,498 (47.23%)	<0.001
Female	4,040,220 (52.64%)	145,300 (49.39%)	3,894,920 (52.77%)	
Age group (years)				
0–5	451,388 (5.88%)	37,801 (12.85%)	413,587 (5.6%)	<0.001
6–17	959,523 (12.5%)	77,720 (26.42%)	881,803 (11.95%)	
18–34	1,733,206 (22.58%)	84,204 (28.62%)	1,649,002 (22.34%)	
35–59	2,728,387 (35.55%)	81,286 (27.63%)	2,647,101 (35.87%)	
60–64	495,728 (6.46%)	6160 (2.09%)	489,568 (6.63%)	
65+	1,306,375 (17.02%)	7018 (2.39%)	1,299,357 (17.61%)	
Education				
Not-completed-education	1,705,508 (22.22%)	102,090 (34.70%)	1,603,418 (21.73%)	<0.001
Less than primary	499,959 (6.51%)	60,581 (20.59%)	439,378 (5.95%)	
Primary	1,416,351 (18.46%)	96,261 (32.72%)	1,320,090 (17.89%)	
Vocational	1,353,740 (17.64%)	25,160 (8.55%)	1,328,580 (18.00%)	
Highschool	2,244,151 (29.24%)	9221 (3.13%)	2,234,930 (30.28%)	
Tertiary	454,898 (5.93%)	876 (0.30%)	454,022 (6.15%)	
Total	7,674,607 (100%)	294,189 (100%)	7,380,418 (100%)	-

**Table 2 ijerph-16-03620-t002:** Crude prevalence of having at least one chronic disease and the prevalence of activity limitations among subjects with chronic disease by functions in Hungary, according to the 2011 census, evaluated by χ^2^-test (more detailed descriptive statistics are summarized in Table A2).

	Total (%)	Ethnicity		Roma / Non-Roma (%/%)	*p*-Value
		Roma (%)	Non-Roma (%)		
Chronic diseases subjects in study population	1,626,447 (21.19%)	43,382 (14.75%)	1,583,065 (21.45%)	0.68	<0.001
Activity limitation among chronic diseased subjects in:					
Self-sufficiency	139,933 (9.88%)	4193 (10.81%)	135,740 (9.85%)	1.09	<0.001
Everyday life	467,747 (33.02%)	16,351 (42.14%)	451,396 (32.78%)	1.28	<0.001
Learning working	218,145 (15.40%)	13,360 (34.43%)	204,785 (14.87%)	2.31	<0.001
Family life	42,022 (2.97%)	2064 (5.32%)	39,958 (2.90%)	1.83	<0.001
Transport	361,757 (25.54%)	10,279 (26.49%)	351,478 (25.51%)	1.03	<0.001
Communication	39,339 (2.78%)	1537 (3.96%)	37,802 (2.74%)	1.44	<0.001
Community life	81,101 (5.73%)	2897 (7.47%)	78,204 (5.68%)	1.31	<0.001

**Table 3 ijerph-16-03620-t003:** Roma ethnicity’s association with the occurrence of having at least one chronic disease and activity limitations among those with chronic disease, according to age- and sex-standardized risk ratios (RR); age-, sex-, and education-standardized risk ratios (RR_e_); and multivariate logistic regression analyses (OR).

	RR [95% CI]	RR_e_ [95% CI]	OR [95% CI]
Chronic disease	1.41 [1.40–1.43]	1.11 [1.10–1.12]	1.17 [1.16–1.18]
Activity limitations in:			
Self-sufficiency	1.75 [1.70–1.80]	1.01 [0.97–1.04]	0.96 [0.93–1.00]
Everyday life	1.64 [1.61–1.66]	1.18 [1.16–1.20]	1.20 [1.17–1.23]
Learning-working	1.59 [1.56–1.62]	1.19 [1.17–1.22]	1.24 [1.21–1.27]
Family life	1.60 [1.53–1.67]	1.22 [1.17–1.27]	1.22 [1.16–1.28]
Transport	1.54 [1.51–1.57]	1.16 [1.14–1.19]	1.03 [1.01–1.06]
Communication	1.44 [1.36–1.51]	0.86 [0.82–0.91]	0.97 [0.92–1.03]
Community life	1.14 [1.10–1.18]	1.03 [0.99–1.07]	1.02 [0.98–1.07]

[95% CI]: 95% confidence interval; OR: odds ratio adjusted for housing conditions (walls quality, public utilities as bathroom, flush toilet, electricity, water and hot water supply, and heating system), and personal characteristics (sex, age, marital status, education, employment).

**Table 4 ijerph-16-03620-t004:** Roma ethnicity’s association with the occurrence of having at least one chronic disease and activity limitations among those with chronic disease, according to age- and sex-standardized risk ratios (RR); age-, sex-, and education-standardized risk ratios (RR_e_); and multivariate logistic regression analyses (OR).

	Attributable Number of Cases among Roma [95% CI]			Attributable Percentage of Cases among Roma [95% CI]			Attributable Percentage of Cases in the Population [95% CI]		
	Age-Sex Adjusted *	Age-Sex-Education Adjusted *	Multiple Adjusted **	Age-Sex Adjusted *	Age-Sex-Education Adjusted *	Multiple Adjusted **	Age-Sex Adjusted *	Age-Sex-Education Adjusted *	Multiple Adjusted **
Chronic diseased subjects in study population	12,615 [12,395; 13,045]	4299 [3944; 4648]	6303 [5984; 6618]	29.08% [28.57; 30.07]	9.91% [9.09; 10.71]	14.53% [13.79; 15.25]	0.78% [0.76; 0.80]	0.26% [0.24; 0.29]	0.39% [0.37; 0.41]
Activity limitation among subjects with chronic disease in:									
Self-sufficiency	1797 [1727; 1864]	42 [−130; 161]	−175 [−316; 0]	42.86% [41.18; 44.44]	0.99% [-3.09; 3.85]	−4.17% [−7.53; 0]	1.28% [1.23; 1.33]	0.03% [−0.09; 0.12]	−0.12% [−0.23; 0]
Everyday life	6381 [6195; 6501]	2494 [2255; 2725]	2725 [2376; 3058]	39.02% [37.89; 39.76]	15.25% [13.79; 16.67]	16.67% [14.53; 18.70]	1.36% [1.32; 1.39]	0.53% [0.48; 0.58]	0.58% [0.51; 0.65]
Learning-working	4957 [4796; 5113]	2133 [1941; 2409]	2586 [2319; 2840]	37.11% [35.90; 38.27]	15.97% [14.53; 18.03]	19.35% [17.36; 21.26]	2.27% [2.20; 2.34]	0.98% [0.89; 1.10]	1.19% [1.06; 1.30]
Family life	774 [715; 828]	372 [300; 439]	372 [285; 452]	37.50% [34.64; 40.12]	18.03% [14.53; 21.26]	18.03% [13.79; 21.88]	1.84% [1.70; 1.97]	0.89% [0.71; 1.04]	0.89% [0.68; 1.07]
Transport	3604 [3472; 3732]	1418 [1262; 1641]	299 [102; 582]	35.06% [33.77; 36.31]	13.79% [12.28; 15.97]	2.91% [0.99; 5.66]	1% [0.96; 1.03]	0.39% [0.35; 0.45]	0.08% [0.03; 0.16]
Communication	470 [407; 519]	−250 [−337; −152]	−48 [−134; 45]	30.56% [26.47; 33.77]	−16.28% [−21.95; −9.89]	−3.09% [−8.7; 2.91]	1.19% [1.03; 1.32]	−0.64%[−0.86; −0.39]	−0.12% [−0.34; 0.11]
Community life	356 [263; 442]	84 [−29; 190]	57 [−59; 190]	12.28% [9.09; 15.25]	2.91% [−1.01; 6.54]	1.96% [−2.04; 6.54]	0.44% [0.32; 0.54]	0.10% [−0.04; 0.23]	0.07% [−0.07; 0.23]

* by standardization; ** by logistic regression model.

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
