# Peer review of "Prevalence of Chronic Diseases and Activity-Limiting Disability among Roma and Non-Roma People: A Cross-Sectional, Census-Based Investigation"

_ijerph, 2019, doi:10.3390/ijerph16193620_

Round 1

Reviewer 1 Report

The manuscript "Prevalence of chronic diseases and activity-limiting disability among Roma and non-Roma: A cross-sectional, census-based investigation" seeks to understand the impact of ethnicity on the occurrence of chronic diseases and limitations on daily life in the Roma population, as well as to explore the possibilities offered by census-based studies to monitor the health of the Roma population and compare it with the non-Roma population. Using the Hungarian national census database and taking self-declaration as the ethnic identifier, the authors concluded that (a) Roma ethnicity is a distinct risk factor for a chronic disease with worse prognosis than the non-Roma population and (b) census-based studies can contribute to improving the health monitoring system of the Roma population.

The article is well written, the study well conducted, as well as the analyses, results and discussion being solid. It is a hot challenge for public health agendas to collect data on Roma ethnic patterns, record it and use it to understand their specific needs and compare them with the rest of the population. The article offers a strategy for using census databases to accomplish this task. The strengths and weaknesses of the methodology are discussed. For that, this paper should be accepted for the special issue “Roma disadvantages”.

As a suggestion, the authors might consider reinforcing their contribution to this special issue with a more extensive discussion of what lessons this study brings regarding the identification of sensitive ethnic indicators of health inequities as experienced by the Roma population, to offer them culturally appropriate services, and to facilitate the clinical management of patients. Beyond the rigorous methodology, data collection in the field of ethnicity is a task that requires extreme sensitivity, and must ensure that this information is used to provide the best services to the population, avoids stigmatization and includes the most vulnerable and silenced groups. The case of the Roma is especially complex because of the history of oppression and stigmatization they have suffered for centuries. The recent report titled "Analysis and comparative review of equality data collection practices in the European Union" and the "Ethnic Minority Health in Ireland (EMH-IC)” project could be only a couple of excellent examples in this endeavour.

Author Response

Dear Reviewer,

Please find enclosed the revised (manuscript ID: ijerph-595957) manuscript „ Prevalence of chronic diseases and activity-limiting disability among Roma and non-Roma people: A cross-sectional, census-based investigation”, by Ferenc Vincze, et al., to be submitted as an article to the International Journal of Environmental Research and Public Health for consideration of publication.

Thank you very much for the careful review of our manuscript. The corresponding changes and refinements made in the revised paper are summarized in our response after considering each of your suggestion. The changes in the manuscript are shown with revision marks.

Answers along with the modifications we made are the following (comments of yours are in capitals):

While re-checking the manuscript we found an error due to rounding. In line 189 the correct frequency of the chronic diseased Roma population is 14.75%. We also corrected this value in the Table 2.

THE ARTICLE IS WELL WRITTEN, THE STUDY WELL CONDUCTED, AS WELL AS THE ANALYSES, RESULTS AND DISCUSSION BEING SOLID. IT IS A HOT CHALLENGE FOR PUBLIC HEALTH AGENDAS TO COLLECT DATA ON ROMA ETHNIC PATTERNS, RECORD IT AND USE IT TO UNDERSTAND THEIR SPECIFIC NEEDS AND COMPARE THEM WITH THE REST OF THE POPULATION. THE ARTICLE OFFERS A STRATEGY FOR USING CENSUS DATABASES TO ACCOMPLISH THIS TASK. THE STRENGTHS AND WEAKNESSES OF THE METHODOLOGY ARE DISCUSSED.

Thank you very much for this positive comment on the topic and manuscript.

AS A SUGGESTION, THE AUTHORS MIGHT CONSIDER REINFORCING THEIR CONTRIBUTION TO THIS SPECIAL ISSUE WITH A MORE EXTENSIVE DISCUSSION OF WHAT LESSONS THIS STUDY BRINGS REGARDING THE IDENTIFICATION OF SENSITIVE ETHNIC INDICATORS OF HEALTH INEQUITIES AS EXPERIENCED BY THE ROMA POPULATION, TO OFFER THEM CULTURALLY APPROPRIATE SERVICES, AND TO FACILITATE THE CLINICAL MANAGEMENT OF PATIENTS. BEYOND THE RIGOROUS METHODOLOGY, DATA COLLECTION IN THE FIELD OF ETHNICITY IS A TASK THAT REQUIRES EXTREME SENSITIVITY, AND MUST ENSURE THAT THIS INFORMATION IS USED TO PROVIDE THE BEST SERVICES TO THE POPULATION, AVOIDS STIGMATIZATION AND INCLUDES THE MOST VULNERABLE AND SILENCED GROUPS. THE CASE OF THE ROMA IS ESPECIALLY COMPLEX BECAUSE OF THE HISTORY OF OPPRESSION AND STIGMATIZATION THEY HAVE SUFFERED FOR CENTURIES. THE RECENT REPORT TITLED "ANALYSIS AND COMPARATIVE REVIEW OF EQUALITY DATA COLLECTION PRACTICES IN THE EUROPEAN UNION" AND THE "ETHNIC MINORITY HEALTH IN IRELAND (EMH-IC)” PROJECT COULD BE ONLY A COUPLE OF EXCELLENT EXAMPLES IN THIS ENDEAVOUR.

Thank you very much for your suggestions. We modified the manuscript with the following statement and the suggested reports: Line 76: “Although there are initiatives to overcome these methodological shortcomings.”

Sincerely yours,

Janos Sandor on behalf of the authors

Reviewer 2 Report

From my read, there are two main ideas in the article.

First, the paper uses the 2011 Hungarian Census data to show the high incidence of chronic diseases and the existence of activity limitations, reaching the conclusion that Roma ethnicity is a risk factor. This is not a new idea, as for years this has been showed by multiple previous studies, surveys conducted ad hoc, qualitative ones etc. Therefore, my main concern is that it does not remain clear which is the new contribution this article is making as it is clear in the available data and the scientific literature that there are these additional risks, this is one of the reason the EC advanced nearly ten years ago the development of the NRIS, in order to tackle the Roma and non-Roma gap in health status. The authors also argue that there is no NRIS monitoring, while FRA (the Fundamental Rights Agency has developed already two rounds 2011 and 2016 of a Roma households survey in order to do so, covering specifically Hungary and health). Authors should support their critiques. 

A second here are some potential in terms of reflecting on the opportunities and limitations that using Census data brings to this type of studies. However, it is not made clear the final conclusions. 

The methodology used and the description of it is correct and appropriate. 

In terms of language, the article is full of long sentences which meaning are  not always are clear. 

Author Response

Dear Reviewer,

Please find enclosed the revised (manuscript ID: ijerph-595957) manuscript „ Prevalence of chronic diseases and activity-limiting disability among Roma and non-Roma people: A cross-sectional, census-based investigation”, by Ferenc Vincze, et al., to be submitted as an article to the International Journal of Environmental Research and Public Health for consideration of publication.

Thank you very much for the careful review of our manuscript. The corresponding changes and refinements made in the revised paper are summarized in our response after considering each of your suggestion. The changes in the manuscript are shown with revision marks.

Answers along with the modifications we made are the following (comments of yours are in capitals):

While re-checking the manuscript we found an error due to rounding. In line 189 the correct frequency of the chronic diseased Roma population is 14.75%. We also corrected this value in the Table 2.

FIRST, THE PAPER USES THE 2011 HUNGARIAN CENSUS DATA TO SHOW THE HIGH INCIDENCE OF CHRONIC DISEASES AND THE EXISTENCE OF ACTIVITY LIMITATIONS, REACHING THE CONCLUSION THAT ROMA ETHNICITY IS A RISK FACTOR. THIS IS NOT A NEW IDEA, AS FOR YEARS THIS HAS BEEN SHOWED BY MULTIPLE PREVIOUS STUDIES, SURVEYS CONDUCTED AD HOC, QUALITATIVE ONES ETC. THEREFORE, MY MAIN CONCERN IS THAT IT DOES NOT REMAIN CLEAR WHICH IS THE NEW CONTRIBUTION THIS ARTICLE IS MAKING AS IT IS CLEAR IN THE AVAILABLE DATA AND THE SCIENTIFIC LITERATURE THAT THERE ARE THESE ADDITIONAL RISKS, THIS IS ONE OF THE REASON THE EC ADVANCED NEARLY TEN YEARS AGO THE DEVELOPMENT OF THE NRIS, IN ORDER TO TACKLE THE ROMA AND NON-ROMA GAP IN HEALTH STATUS. THE AUTHORS ALSO ARGUE THAT THERE IS NO NRIS MONITORING, WHILE FRA (THE FUNDAMENTAL RIGHTS AGENCY HAS DEVELOPED ALREADY TWO ROUNDS 2011 AND 2016 OF A ROMA HOUSEHOLDS SURVEY IN ORDER TO DO SO, COVERING SPECIFICALLY HUNGARY AND HEALTH). AUTHORS SHOULD SUPPORT THEIR CRITIQUES.

The Second European Union Minorities and Discrimination Survey study conducted by the European Union Agency for Fundamental Rights reached a 56% response rate in the 16 or years older sampled population. Representative conclusions can only be drawn for those areas where density of Roma population higher than 10%. A census based investigation covers the whole Hungarian population without sampling error in a situation so-called ‘hard-to-reach’ group. According to the suggestion, in Line 81-83 we made the following extension of argumentation: “Taking into account that surveys are inevitably subject to sampling bias due to the nonexistence of a Roma sampling frame (e.g. the recently conducted Second European Union Minorities and Discrimination Survey study coordinated by the European Union Agency for Fundamental Rights has representativity limitations arising from the sampling frame) [36], a census that would provide data representative of the population could be a useful part of monitoring.”

A SECOND HERE ARE SOME POTENTIAL IN TERMS OF REFLECTING ON THE OPPORTUNITIES AND LIMITATIONS THAT USING CENSUS DATA BRINGS TO THIS TYPE OF STUDIES. HOWEVER, IT IS NOT MADE CLEAR THE FINAL CONCLUSIONS.

We highlighted and confirmed the manuscript’s most important finding in the “Strengths and Limitations” section by inserting the “The main strength of this” sentences, and in the “Conclusion” section with the “Census based impact assessment of Roma ethnicity on chronic disease occurrence of 1.16% was not described formerly neither by analytic investigations nor households surveys.” statement.

THE METHODOLOGY USED AND THE DESCRIPTION OF IT IS CORRECT AND APPROPRIATE.

Thank you very much for this positive comment on the methodology.

IN TERMS OF LANGUAGE, THE ARTICLE IS FULL OF LONG SENTENCES WHICH MEANING ARE NOT ALWAYS ARE CLEAR.

We re-checked the manuscript, and reduced the number of the long sentences. We hope this will make the text more clear than before.

The following modifications were carried out:

line 47:

Original version: …unhealthier lifestyle than the non-Roma population [5–10]; in addition, the access of Roma people to care is impaired, leading to high levels of unmet health needs [4,11–18].

Corrected version: …unhealthy environments and have an unhealthier lifestyle than the non-Roma population [5–10]. In addition, the access of Roma people to care is impaired, leading to high levels of unmet health needs [4,11–18].

line 68:

Original version: …National Roma Integration Strategies up to 2020” (NRIS) in 2011, which commits Member States to integrate policies that focus on Roma minorities in a “clear and specific” way …

Corrected version: …National Roma Integration Strategies up to 2020” (NRIS) in 2011. It commits Member States to integrate policies that focus on Roma minorities in a “clear and specific” way…

line 115:

Original version: …characterized by these parameters, which were used as proxy indicators of deprivation…

Corrected version: …characterized by these parameters. All factors were used as proxy indicators of deprivation…

line 172:

Original version: … limitation among those with chronic disease was 87.09%, which resulted in a database of 1,416,424 persons with chronic disease…

Corrected version: …limitation among those with chronic disease was 87.09%. This resulted in a database of 1,416,424 persons with chronic disease…

line 242:

Original version: …demonstrating that Roma people have an elevated risk both for chronic disease occurrence and for each studied activity limitation, suggesting that both disease development and…

Corrected version: …demonstrating that Roma people have an elevated risk both for chronic disease occurrence and for each studied activity limitation. These results suggest that both disease development and…

line 249:

Original version: …learning-working, family life, and transport, while in the field of communication, this risk proved to be reduced among Roma people…

Corrected version: …learning-working, family life, and transport. While in the field of communication, this risk proved to be reduced among Roma people…

line 321:

Original version: …among Roma can be identified by the census, while the details of such interventions can be ensured

Corrected version:…among Roma can be identified by the census. While the details of such interventions can be ensured…

line 329:

Original version: …ethnicity is a distinct risk factor independent of sociodemographic status, with significant impact among the Roma population

Corrected version: …ethnicity is a distinct risk factor independent of sociodemographic status.  It suggests significant impact among the Roma population…

Sincerely yours,

Janos Sandor on behalf of the authors
